# Compound Fault Feature Extraction of Rolling Bearing Acoustic Signals Based on AVMD-IMVO-MCKD

**DOI:** 10.3390/s22186769

**Published:** 2022-09-07

**Authors:** Shishuai Wu, Jun Zhou, Tao Liu

**Affiliations:** Faculty of Mechanical and Electrical Engineering, Kunming University of Science and Technology, Kunming 650500, China

**Keywords:** adaptive variational mode decomposition, improved multiverse optimization algorithm, maximum correlated kurtosis deconvolution, bearing compound fault, acoustic diagnosis

## Abstract

The compound fault acoustic signal of a rolling bearing has the characteristics of a varying noise mixture, a low signal-to-noise ratio (SNR), and nonlinearity, which makes it difficult to separate and extract exactly the fault features of compound fault signals. A fault feature extraction approach combining adaptive variational modal decomposition (AVMD) and improved multiverse optimization (IMVO) algorithm parameterized maximum correlated kurtosis deconvolution (MCKD)—named AVMD-IMVO-MCKD—is proposed. In order to adaptively select the parameters of VMD and MCKD, an adaptive optimization method of VMD is proposed, and an improved multiverse optimization (IMVO) algorithm is proposed to determine the parameters of MCKD. Firstly, the acoustic signal of bearing compound faults is decomposed by AVMD to generate several modal components, and the optimal modal component is selected as the reconstruction signal depending on the minimum information entropy of the modal components. Secondly, IMVO is utilized to select the parameters of MCKD, and then MCKD processing is performed on the reconstructed signal. Finally, the compound fault features of the bearing are extracted by the envelope spectrum. Both simulation analysis and acoustic signal experimental data analysis show that the proposed approach can efficiently extract the acoustic signal fault features of bearing compound faults.

## 1. Introduction

The rolling bearing is an important part of a rotating machine, the failure of which will cause the failure of other parts of the rotating equipment [1]. The investigation results show that rolling bearing failures account for 30% of all machine failures [2]. In actual industrial sites, the working conditions of bearings are very harsh, such as noise interference, high temperatures, and strong corrosion conditions. In practical engineering applications, rolling bearings are often mixed with two or more kinds of faults [3]. As a consequence, the compound fault diagnosis of the rolling bearing is more difficult [4,5,6,7].

In recent years, the work of detecting and diagnosing bearing faults based on vibration signal methods has achieved rich results. At the same time, the diagnosis of bearing fault defects by acoustic methods has also been paid more attention [8,9]. A method based on the combination of acoustic emission and vibration signal analysis [10,11] overcomes the shortcoming that vibration analysis cannot detect the early faults of bearings. Yoon et al. [12] proposed acoustic emission technology to realize bearing fault diagnosis at low speeds. The acoustic signal adopts a non-contact measurement method, which not only makes it easy to collect the acoustic signal but also is not limited by the working environment conditions of bearings [13]. Many scholars have made great achievements in the detection of rolling bearing failures by acoustic methods. In the field of deep learning, Liu et al. [14] used short-time Fourier transform (STFT) to preprocess the acoustic signal, and the fault features of the rolling bearing were extracted through stacked sparse autoencoders. Zhang et al. [15] successfully applied deep graph convolutional networks to bearing acoustic fault diagnosis. Kim et al. [16] studied bearing fault diagnosis combining acoustic emission technology with a convolutional neural network. There have been many achievements in the deep learning research neighborhood of acoustic signal analysis methods for rolling bearings [17,18]. In machine acoustics, research based on machine learning has also made progress [19,20,21]. Verma et al. [22] designed a fault condition monitoring system for reciprocating air compressors, and realized fault diagnosis by analyzing the collected acoustic signals. In addition, acoustic signal analysis has also achieved many research results in the field of blind source separation (BSS) [23,24]. Qin et al. [25] introduced an acoustic signal compound fault detection based on the combination of improved empirical wavelet transform (IEWT) and singular value decomposition (SVD). All in all, the fault diagnosis of a bearing based on acoustic signals has been paid more and more attention by scholars.

During the research on compound fault diagnosis of bearings, the adaptive decomposition method has become more popular [26]. Jena et al. [27] analyzed the vibration and acoustic signals of gear faults and bearing faults in order to affirm the availability of the filter system. Amarnath and Krishna [28] applied the empirical mode decomposition (EMD) algorithm to the fault detection of the sound signals of bearings and helical gears. Aiming at the problem of EMD endpoint effect and modal mixing, many scholars have proposed improved methods of EMD. As illustrations, the ensemble empirical mode decomposition (EEMD) method [29], the complete ensemble empirical mode decomposition with adaptive noise (CEEMDAN) method [30], and the improved complete ensemble empirical mode decomposition with adaptive noise (ICEEMDAN) method [31] are listed in this paper. Yang et al. [32] used an EEMD algorithm to preprocess acoustic signals. Lei et al. [33] applied the CEEMDAN algorithm to study bearing faults. In order to address the modal mixing defect, Zheng et al. [34] presented a novel partly EEMD technique. In addition, the researchers have also developed a new adaptive decomposition algorithm, variational mode decomposition (VMD) [35], which has a rigorous mathematical derivation. The penalty factor and modal number of VMD must be predetermined, unlike the approaches mentioned above. For the parameter determination of algorithms such as VMD and maximum correlation kurtosis deconvolution (MCKD), the optimization algorithm can be considered first [36], but the calculation cost of the optimization algorithm is high; the second consideration is to study other algorithms in order to determine the parameters. For example, Li et al. [37] proposed information entropy to determine the parameters of VMD. Yin et al. [38] introduced a new method of relative entropy—which is also known as Kullback-Leibler (KL) divergence—to optimize VMD, which was named KL-VMD. Research [39,40] shows that VMD cannot completely extract fault feature information under strong noise interference. Compared with the feature extraction of single faults, compound fault diagnosis is more difficult, especially for the sound signal analysis of compound faults. Therefore, a new method combining AVMD with MCKD is proposed, which takes full advantage of VMD and MCKD to diagnose the compound faults of rolling bearings. Firstly, we propose the adaptive optimization of VMD based on relative entropy and information entropy, and screen out the penalty factor and modal number according to the minimum sum of the two entropies. The optimal component of VMD containing abundant fault information is screened out by the minimum value of information entropy. Secondly, the order of the shift and the filter length of MCKD are adaptively determined by the improved multiverse optimization (IMVO) algorithm, and MCKD processing is performed on the selected intrinsic mode functions (IMFs). Finally, the envelope spectrum of the optimal modal component processed by MCKD is obtained, and the fault frequency is extracted by the envelope spectrum. The main contributions of this work are as follows:(1)A new method based on AVMD-MVO-MCKD is proposed to extract the features of composite faults accurately.(2)Adaptive variational mode decomposition is proposed. Compared with the traditional signal decomposition method, the proposed method not only overcomes the shortcoming of selecting parameters based on empirical knowledge but also solves the mode aliasing problem. Compared with the existing optimization algorithms, the computational complexity is reduced.(3)The IMVO algorithm is proposed to optimize the MCKD parameters. The computational efficiency of the IMVO algorithm is improved, and the important parameters of MCKD are determined adaptively.(4)The proposed method has been successfully applied to the field of compound fault acoustic signals, which has certain reference value for further research of acoustic signal diagnosis methods.

The remainder of the thesis is structured as follows. The theory of the proposed method is introduced in Section 2. In Section 2.1 and Section 2.2, VMD theory is briefly introduced and VMD parameter optimization based on relative entropy and information entropy is detailed. The theory of the improved multiverse optimization (IMVO) algorithm is detailed in Section 2.3. The brief introduction of MCKD theory and the detailed steps of the IMVO algorithm to adaptively determine MCKD parameters are presented in Section 2.4 and Section 2.5, respectively. In Section 3, the feasibility of the proposed approach is proven by simulation data analysis, and the analysis effects of the proposed method and the comparison method are shown. In Section 4, the analysis of the measured data, the application of the proposed algorithm, and the demonstration of the analysis results of the comparative method are described. Section 5 is our conclusion.

## 2. Proposed Method

### 2.1. VMD Theory

Decomposing an input original signal into several IMFs is the core function of VMD. The VMD needs to settle the important parameter penalty factor and modal number before analyzing the input signal. After the parameters are determined in advance, the input signal is decomposed by the VMD into multiple IMFs. VMD addresses the shortcomings of the EMD algorithm lacking mathematical theory and modal mixing. The theory of VMD will not be carefully introduced here because of space limitations. Detailed mathematical theory derivation can be found in the literature [35].

### 2.2. Determination of the Penalty Factor and Modal Number of VMD

For the crucial parameter presetting problem of VMD, a new parameter optimization method based on the minimum sum of information entropy and relative entropy is proposed in order to select the parameters of VMD adaptively. The minimum sum of information entropy and relative entropy is calculated as an indicator of iteration termination. The calculation process is as follows:(1)$$E(P,Q)=H(Q)+\left|D_{KL}(Q/P)\right|$$
where H(Q) represents the information entropy of the modal component series Q. $D_{KL}(Q/P)$ is the sum of the relative entropy of the original signal P and a series of IMFs Q, which is defined as Formula (2)
(2)$$D_{KL}(Q/P)=\sum_{i=1}^{N}Q(x)\log(\frac{Q(x)}{P(x)})$$
where P(x) is defined as the original input signal, and Q(x) is the IMFs obtained after decomposing the original input signal. The parameters of VMD are determined according to the minimum value of the sum of information entropy and relative entropy. The important parameter optimization process of VMD is presented in Figure 1. Firstly, the search range of mode number K is set [2, 16], and the search step size is 1. The number of modes K = 2 is initialized, and the penalty factor α=2000 is set. The VMD is used to decompose the original signal, and the sum of information entropy and relative entropy is calculated from Equation (1). We then ask whether it is the minimum value of entropy within the search range. If the judgment conditions are met, the optimal modal value is output; if not, then K = K + 1 continues the iterative calculation. Next, the penalty factor α is optimized based on the minimum entropy judgment condition in Equation (1). The search range of penalty factor α is set [100, 2000], and the search step size is 50. We then ask whether it is the minimum entropy in the search range. If the judgment conditions are met, the best penalty factor is output; if not, α=α+50 continues the iterative calculation. Finally, the optimal number of modes K and penalty factor α are output.

### 2.3. Improved Multiverse Optimization (IMVO)

In 2015, Mirjalili et al. [41] proposed the Multiverse Optimization (MVO) algorithm, which is inspired by the black hole, white hole and wormhole in the multiverse theory. Every universe has an inflation rate. A universe with a high inflation rate tends to produce white holes, and conversely black holes appear. In order to determine the best location in the search space, the algorithm transports things through the carrier wormhole from the white hole in the source universe to the black hole in the destination universe in line with cosmic laws. The search process of the MVO algorithm is divided into two phases: exploration and development. White holes and black holes act on the exploration stage, while wormholes act on the development phase. The theoretical derivation of the algorithm is as follows.

Suppose the following search space exists in the universe matrix:(3)$$U=\left[\begin{array}{cccc} x_{1}^{1} & x_{1}^{2} & \cdots & x_{1}^{d}\\ x_{2}^{1} & x_{2}^{2} & \cdots & x_{2}^{d}\\ \vdots & \vdots & \vdots & \vdots \\ x_{n}^{1} & x_{n}^{2} & \cdots & x_{n}^{d} \end{array}\right]$$

In the formula, d means the number of variables, and n means the number of universes.

The search process of the algorithm is carried out under the roulette wheel mechanism, in which each iteration needs to select a white hole.
(4)$$x_{i}^{j}=\left\{\begin{array}{cc} x_{k}^{j}, & r_{1}<NI\left(X_{i}\right)\\ x_{i}^{j}, & r_{1}\geq NI\left(X_{i}\right) \end{array}\right.$$
where $x_{i}^{j}$ is the jth number of the ith universe, $NI\left(X_{i}\right)$ represents the standard expansion rate of the ith universe, $r_{1}$ represents a random variable between [0, 1], and $x_{k}^{j}$ is the jth variable of kth universes determined according to the roulette wheel mechanism.

In the iteration process, the wormhole existence probability (WEP) and the travel distance rate (TDR) are crucial variables of the algorithm. For the purpose of improving the efficiency of the optimization process, the linear growth of WEP is changed to logarithmic growth in Ref. [42]. The TDR becomes continually larger during the iteration, and this change is mainly intended to improve the accuracy of the local search ability. The modified adaptive formulas are Equations (5) and (6).
(5)$$WEP=\ln^{e^{WEP_{\min}}+(e^{WEP_{\max}}-e^{WEP_{\min}})\times \frac{1}{H}}$$
(6)$$TDR={\left(\frac{1}{Q}\right)}^{\frac{l}{H}}\times 0.6$$
where $WEP_{\min}=$ 0.2, $WEP_{\max}=$1, l defines the current iteration, H indicates the maximum iterations, and Q= 5000.

The location of the universe is searched, and the optimal position is found according to Equation (7).

When $r_{2}<WEP$,
(7)$$x_{i}^{j}=\left\{\begin{array}{cc} x_{j}+T_{DR}\times \left(\left(ub_{j}-lb_{j}\right)\times r_{4}+lb_{j}\right), & r_{3}<0.5\\ x_{j}-T_{DR}\times \left(\left(ub_{j}-lb_{j}\right)\times r_{4}+lb_{j}\right), & r_{3}\geq 0.5 \end{array}\right.$$

When $r_{2}\geq WEP$,
(8)$$x_{i}^{j}=x_{i}^{j}$$

In the formula, $x_{j}$ represents the jth parameter of the current optimal universe. $ub_{j}$ and $lb_{j}$ represent the upper and lower bounds of the jth parameter. $r_{2}$, $r_{3}$ and $r_{4}$ are random variables with values between 0 and 1.

### 2.4. MCKD Theory

MCKD obtains the maximum correlation kurtosis of the original input signal by selecting the optimal FIR filter, such that the output signal can recover the characteristics of the original signal. For the selection of the filter length and the order of shift, which are important parameters of MCKD, an improved multiverse optimization (IMVO) algorithm is used to select them adaptively. The theoretical derivation details of the MCKD algorithm were introduced in the Ref. [43].

### 2.5. IMVO-Optimized MCKD Parameters

Two crucial parameters for MCKD are optimized and selected by the IMVO algorithm. The steps of IMVO to optimize MCKD parameters are as follows:(1)In the process of parameter setting, the maximum number of iterations H of IMVO is 50, and the number of universes n is 30; the optimization range of filter length L of MCKD is set to [100, 300], and the order of shift M is set to [1, 7]. According to the formula $T=f_{s}/f_{m}$, the inner ring convolution period $T_{i}$ and the outer-ring deconvolution period $T_{o}$ are calculated. This initializes the location of the universe randomly based on the parameter setting range. (2)The inflation rates of the universe are taken and ranked, then a white hole is selected under the roulette wheel mechanism.(3)The WEP and TDR are updated according to Equations (5) and (6), and then boundary checking is performed.(4)We then perform a calculation operation on the current inflation rate of the universe. While the cosmic inflation rate is better than the current cosmic expansion rate, the current cosmic expansion rate is updated; otherwise, we keep the current cosmic expansion rate.(5)The universe position update is executed, and the optimal individual is found according to Formula (7).(6)When judging the termination condition, if the termination condition is satisfied, the iteration will be terminated, and the result will be output. If not, the iteration will be continued by returning to step (2).

To sum up, the specific steps of the proposed method are presented in Figure 2.

## 3. Simulation Analysis

The feasibility of the improved algorithm is proven by the simulation of mixed signals. The mathematical expression [44] of the bearing inner-ring fault is as follows:(9)$$x(t)=\sum_{i=1}^{M}A_{i}s(t-iT-\tau_{i})+n(t)$$
(10)$$A_{i}=A_{0}\cos(2\pi Qt+\varphi_{A})+C_{A}$$
(11)$$s(t)=e^{-Bt}\sin(2\pi f_{n}t+\varphi_{\omega })$$

In the formula, the natural frequency is 3000 Hz, the sampling frequency is 8192 Hz, and the system attenuation coefficient is 500.

Formula (12) shows the mathematical model expression of the bearing outer-ring fault [45]:(12)$$y_{o}=A_{1}e^{\left(-2g\pi f_{n}t_{0}\right)}\cdot \sin\left(2\pi f_{n}\sqrt{1-g^{2}}\cdot t_{0}\right)$$
where *g* defines the damping coefficient, and $t_{0}$ is the single-cycle sampling time.

The random mixtures of the inner-ring fault simulation signal, the outer-ring fault simulation signal and Gaussian white noise with a signal-to-noise ratio (SNR) of −8 dB are operated by computer, and then the mixed signal is obtained. Table 1 presents the detailed parameters of the mathematical model expressions of bearing defects. Figure 3a shows the time-domain waveforms of the inner-ring fault simulation signal, the outer-ring fault simulation signal, and the mixed signal, respectively; Figure 3b represents the envelope spectrum of the inner-ring fault simulation signal, the outer-ring fault simulation signal, and the mixed signal, respectively.

In Figure 3a, the bearing fault impact component of the mixed signal is obviously submerged by noise, and no valuable fault message can be obtained. The failure to obtain valuable fault feature information in the envelope spectrum of the mixed signal in Figure 3b indicates that it is severely affected by noise. The adaptive VMD (AVMD) decomposition is performed on the mixed signal. Figure 4 displays the decomposition results of the AVMD. With the method introduced in Section 2.2, the parameter combination of VMD which is adaptively obtained is $\left[K,\alpha \right]=$ [3, 100]. The information entropy value of each IMF component is obtained by relying on the information entropy formula, and the information entropy value of each IMF component is filled in Table 2. The IMF component corresponding to the minimum information entropy value is picked as the optimal component from Table 2. Because IMF3 has the smallest information entropy value, IMF3 is chosen as the optimal component.

Figure 5 is the envelope spectrum analysis of IMF3 selected by information entropy. From Figure 5, it can be observed that the fault features of the inner ring and outer ring are mixed with each other, which means that bearing compound fault features cannot be separated. Therefore, MCKD is used to analyze IMF3 to accomplish the goal of separating fault features. The deconvolution period is obtained from Equation (13).
(13)$$T=f_{s}/f_{m}$$

In the equation, $f_{s}$ defines the sampling frequency, and $f_{m}$ indicates the fault frequency.

After calculation, the deconvolution period of the inner-ring fault signal is $T_{i}=$ 86.2, and the range of parameter $T_{i}$ is set [85, 89]; the deconvolution period of the outer-ring fault signal is $T_{o}=$ 128, and the range of parameter $T_{o}$ is set to [126, 130]. After analyzing massive simulation experiments, it is determined that the parameter combination of MCKD is $\left[T_{i}=85,T_{o}=128\right]$. The parameter filter length L and the order of shift M of MCKD are adaptively selected by the IMVO algorithm. After IMVO optimization, the parameter combination of MCKD is determined adaptively as $\left[L,M\right]=$ [287, 1].

Figure 6 is the time-domain waveform of the optimal IMF3 after MCKD processing. Figure 7 represents the envelope spectrum of the optimal IMF3 after MCKD processing. In Figure 6, the impact component of the signal is obvious, which shows that MCKD has accomplished the purpose of reducing the noise and recovering the impact component. The fault features of the inner and outer rings of the mixed signal are fully separated and extracted in Figure 7. The fault frequency and harmonics of the inner ring are fully obtained, and the fault frequency of the outer ring is clearly presented in Figure 7. In order to illustrate the superiority of the algorithm in the paper, KL-VMD-MVO-MCKD, EEMD-MVO-MCKD and ICEEMDAN-MVO-MCKD methods are selected as comparative experiments. Figure 8 represents the envelope spectrum of a mixed signal processed using KL-VMD-MVO-MCKD. Figure 9 represents the envelope spectrum of a mixed signal processed by using EEMD-MVO-MCKD. Figure 10 represents the envelope spectrum of a mixed signal processed by using ICEEMDAN-MVO-MCKD. Although the inner-ring fault frequency can be extracted, it is still mixed with the outer-ring fault frequency $f_{o}$; the outer-ring fault frequencies $f_{o}$ and $3f_{o}$ can be obtained, but the amplitude is not obvious in Figure 8. The fault frequencies of inner and outer rings are separated and extracted, but the harmonics are not obtained in Figure 9 and Figure 10. The results of the comparison method KL-VMD-MVO-MCKD are not as good as the method proposed in this paper, and the calculation efficiency of the comparison method is low. The interference components around the fault frequency spectrum lines are obvious in Figure 9 and Figure 10. From the spectral lines of Figure 9 and Figure 10, it can be seen that the noise reduction ability of the two algorithms in the comparative experiment is not as good as the proposed algorithm.

From the spectral lines in Figure 7, it can be concluded that not only can the fault features of inner and outer rings be separated and extracted but also harmonics can be obtained. The spectral lines of the inner-ring fault frequency and the outer-ring fault frequency are very clear in Figure 7. In a word, the analysis effects of the three comparison methods are not as good as the method proposed in this paper. Therefore, the proposed method exhibits the advantages of a good noise reduction effect, a high accuracy of fault feature separation and extraction, and high computation efficiency. The analysis consequences of the simulated mixed signal confirm the feasibility of the proposed approach, which is applicative for the compound fault feature extraction of rolling bearing acoustic signals under a noise environment.

## 4. Experimental Analysis

The rolling bearing in the experiment includes inner-ring failure and outer-ring failure. The measured data of compound rolling bearing faults were utilized to test the method in this paper. The measured data came from the laboratory, and two Beijing prestige sensors were used to pick up the sound signals of bearing compound faults when the experimental bench built by the QPZZ-II rotating equipment is working. The layout of the test bench and microphone is presented in Figure 11. Figure 11 shows that microphone 1 was placed in the flush position of the centerline of the side of faulty bearing housing, and microphone 2 was installed in the aligned position of the main shaft of bearing, the two microphones were placed perpendicular to each other and installed at a position of 0.5m from the edge of the faulty bearing seat and 0.5 m in height. The NI Signal Express acquisition module and NI-9234 four-channel acquisition card were used to acquire the fault signal. The sampling frequency was 8192 Hz, and the sampling length of the measured data was 8192. The bearing model used in the experiment was NU205, and its technical parameters are presented in Table 3. The inner-ring defect and outer-ring defect of the bearing were obtained by wire cutting. With the bearing parameters presented in Table 3, the following information can be obtained. Through calculation, the fault frequency of the inner ring was 95.38 Hz, and that of the outer ring was 64.61 Hz. The rotation speed in the experiment was set to 800 r/min, that is, the rotational frequency $f_{r}$ was 13.33 Hz.

The fault type of rolling bearing is shown in Figure 12. Figure 12a is the inner-ring defect, Figure 12b is the outer-ring defect. Figure 13 presents the time-domain waveform and envelope spectrogram of the original acoustic dataset picked up by the two microphones. The mixed noise in Figure 13a is obvious. There is very little useful information obtained from Figure 13b, which can only preliminarily judge that the inner- and outer-ring fault frequency of the bearing are mixed with each other. Because the envelope spectrum of the acoustic signal collected by microphone 2 cannot display useful information, the second signal was selected for AVMD decomposition, and the decomposition consequences are presented in Figure 14. The parameter combination $\left[K,\alpha \right]=$ [3, 100] of VMD is adaptively determined by the improved algorithm proposed in Section 2.2. The information entropy of each IMF component is filled in Table 4. Table 4 reveals that the IMF3 information entropy of the measured acoustic signal is the smallest, and IMF3 is determined to be the optimal component. The spectral lines of the envelope of IMF3 are presented in Figure 15. The inner- and outer-ring fault features of rolling bearings are mixed with each other, which means that the compound fault features of the bearing cannot be separated in Figure 15. Therefore, the optimal components were analyzed by MCKD to separate the different fault information. After calculation, the inner-ring deconvolution period is $T_{i}$ = 85.89, and the outer-ring deconvolution period is $T_{o}=$ 126.79. Through many experimental analyses, it is determined that the inner-ring deconvolution period and outer-ring deconvolution period of the MCKD are [T = 86, T = 126], respectively. The parameters L and M of MCKD are adaptively selected by the IMVO algorithm, and the parameter combination is $\left[L,M\right]=$ [269, 1].

Figure 16 and Figure 17, respectively, represent the time-domain waveform and envelope spectrum of the optimal component IMF3 after MCKD processing. The shock component of the signal in Figure 16 is obvious, which shows that the MCKD algorithm has superior performance. The mixed inner-ring and outer-ring fault features in Figure 17 are sufficiently separated and extracted. The KL-VMD-MVO-MCKD, EEMD-MVO-MCKD and ICEEMDAN-MVO-MCKD methods are selected as comparative experiments to prove the superiority of the proposed approach. Figure 18 is the envelope spectrum of an acoustic signal processed using KL-VMD-MVO-MCKD. Figure 19 is the envelope spectrum of an acoustic signal processed using EEMD-MVO-MCKD. Figure 20 is the envelope spectrum of the acoustic signal analyzed using ICEEMDAN-MVO-MCKD. The fault frequencies of the inner and outer rings are clearly separated and extracted, and the harmonics are obtained in Figure 17. Although part of the outer-ring fault frequency and inner-ring fault frequency can be obtained from the envelope spectrum, the spectral line amplitude representing the fault frequency is not obvious in Figure 18. The inner-ring fault information cannot be easily obtained from the spectral lines, and the outer-ring fault information can only be obtained with fewer fault frequencies such as $f_{o}$ and $2f_{o}$ from the spectral lines in Figure 19. It can be observed from Figure 20 that the fault information of the inner ring cannot be obtained, and most of the fault frequencies of the outer ring can only be obviously separated and extracted.

The comparative experimental analysis results show that the accuracy of fault feature separation and extraction by KL-VMD-MVO-MCKD is low and the computational efficiency is low. The experimental results show that the performance of EEMD-MVO-MCKD and ICEEMDAN-MVO-MCKD in the separation and extraction of compound fault features is insufficient, and the noise reduction effect is not good enough. However, the approach proposed in the paper not only separates and extracts the fault features of inner and outer rings but also works well. Therefore, the proposed algorithm not only resolves the shortcomings of traditional decomposition methods that the IMF components contain residual noise and the IMF components only contain signal local fault features but also improves the accuracy of the separation and extraction of compound fault features. Meanwhile, the computational complexity of the proposed algorithm is reduced, and the computational efficiency is improved. In this paper, the method of using AVMD-IMVO-MCKD to detect the acoustic signal of a bearing compound fault is feasible and efficient. The effects of the comparative experiments also verify the applicability and efficiency of the proposed algorithm in the paper.

## 5. Conclusions

In this paper, a new feature extraction method based on AVMD-IMVO-MCKD was proposed for rolling bearing compound fault acoustic signals. The main innovations were in two aspects: firstly, we proposed to adaptively determine the parameters of VMD by relying on the minimum entropy as the criterion. The proposed method overcomes the shortcoming of selecting parameters manually by relying on rich experience, and the computational efficiency of the proposed algorithm is very high compared with the optimization algorithm. Compared with traditional signal decomposition methods, AVMD solves the problems of mode aliasing and insufficient decomposition ability. It only takes an average of 3 min to determine the modal number and the penalty factor. Secondly, an improved multiverse optimization (IMVO) algorithm was proposed to adaptively obtain the critical parameters of MCKD. The IMVO algorithm improves the computational efficiency and reduces the computational complexity. The superiority of the proposed method is illustrated by the comparative analysis. The results show that the combination of VMD and MCKD is very necessary. Compared with the existing methods, the proposed approach in the paper can fully separate and extract the fault features of acoustic signal compound faults. However, the deconvolution period parameter of MCKD is obtained according to the formula in the paper, and the optimization algorithm can be considered to determine the deconvolution period in the future.

## Figures and Tables

**Figure 1 sensors-22-06769-f001:**
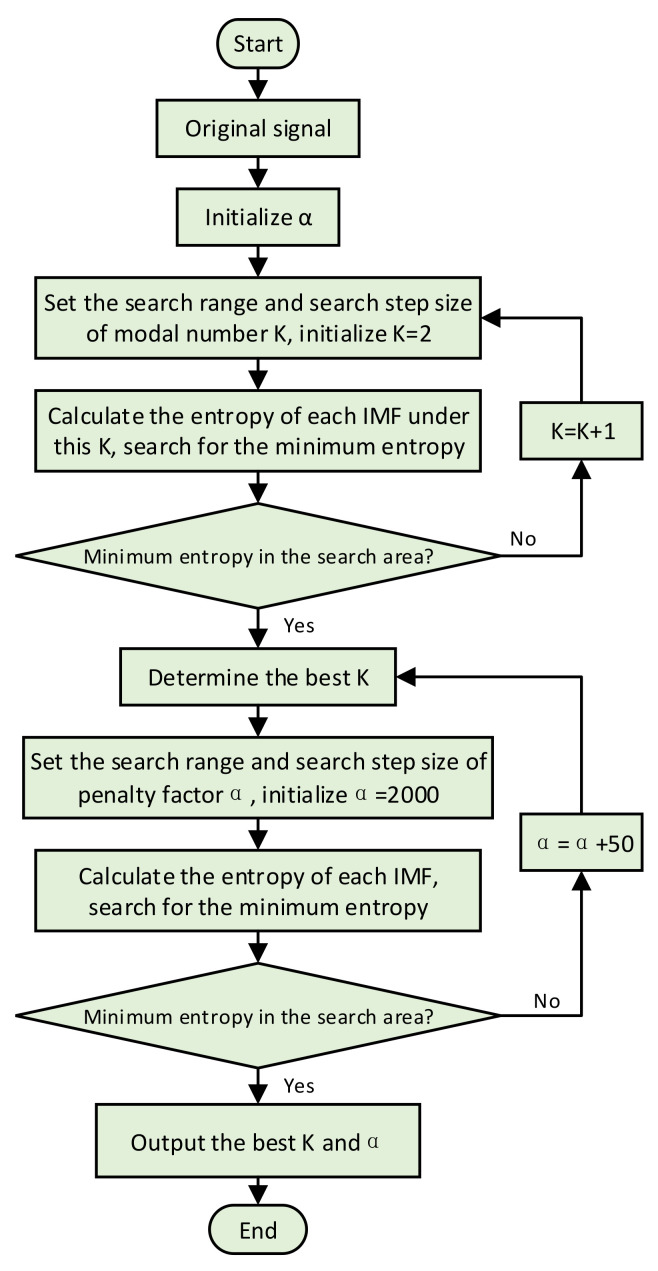
VMD parameter adaptive selection flow chart.

**Figure 2 sensors-22-06769-f002:**
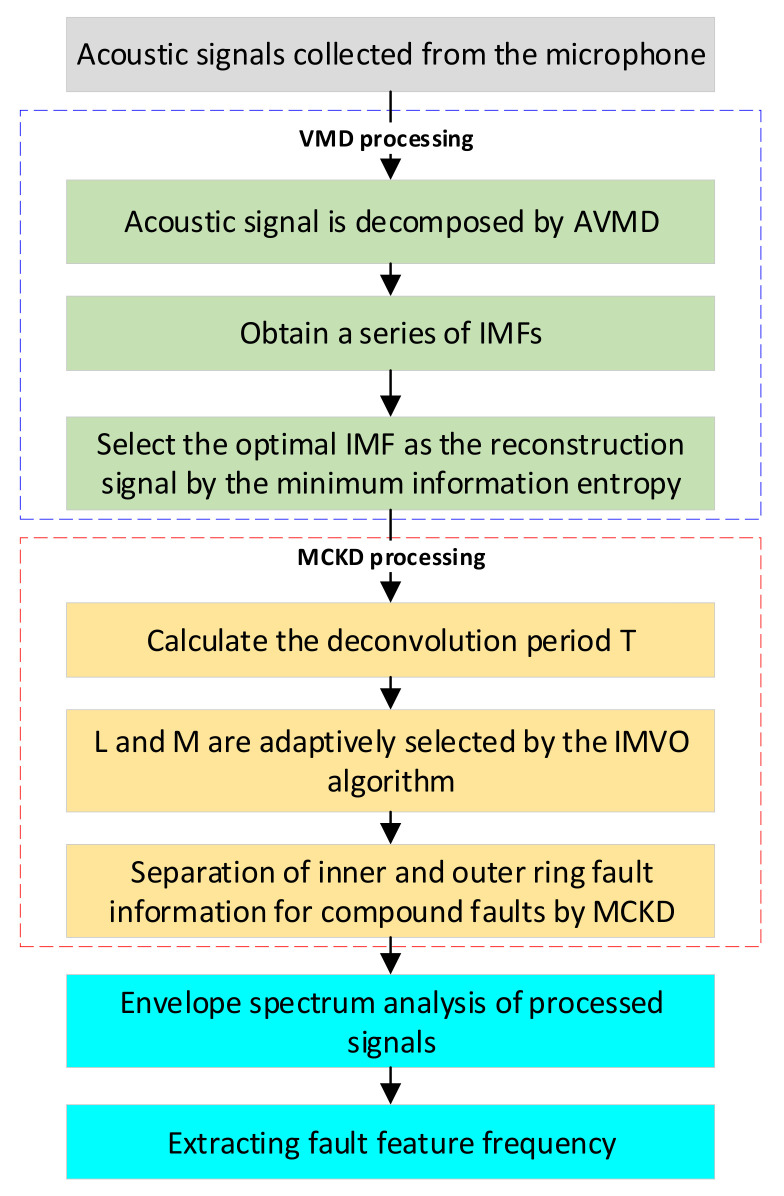
The detailed steps of AVMD-IMVO-MCKD for an acoustic signal.

**Figure 3 sensors-22-06769-f003:**
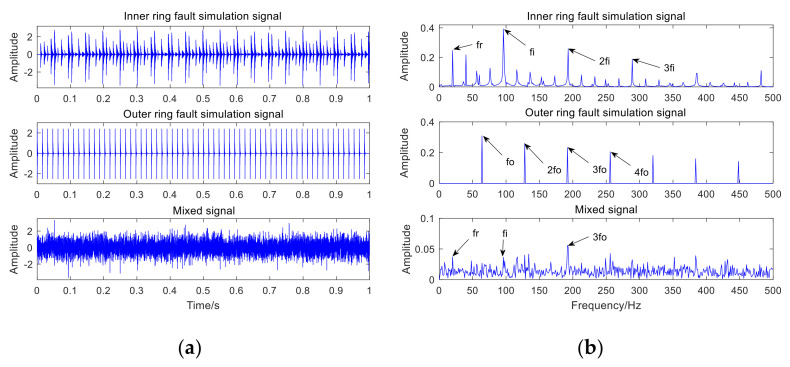
Time and spectral domain results of simulation signals: (**a**) time-domain waveform of the simulation data; (**b**) envelope spectrum of the simulation data.

**Figure 4 sensors-22-06769-f004:**
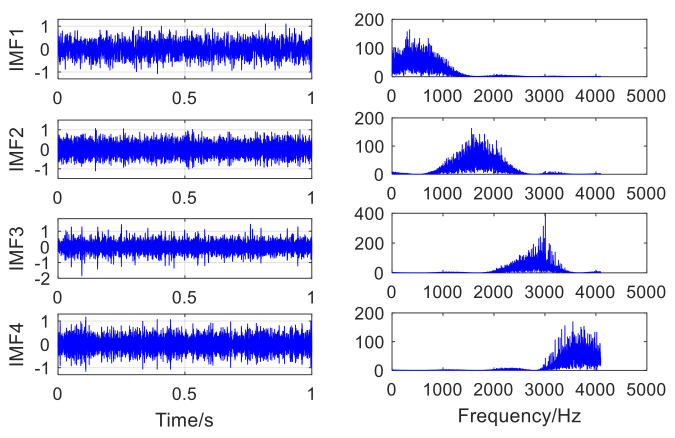
The result of decomposing the mixed signal by AVMD.

**Figure 5 sensors-22-06769-f005:**
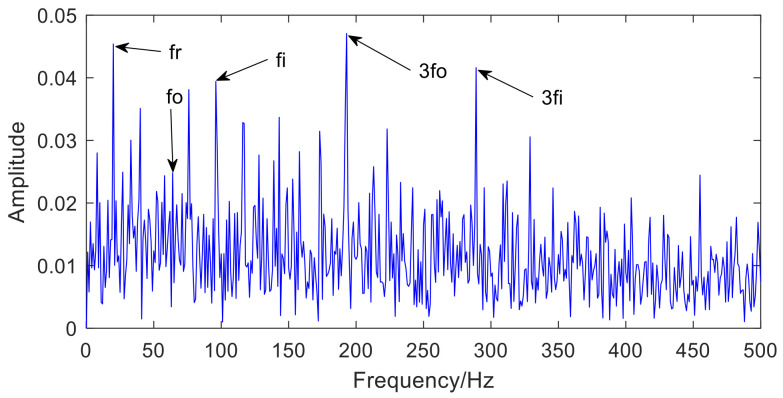
The envelope spectrum of IMF3 selected by information entropy.

**Figure 6 sensors-22-06769-f006:**
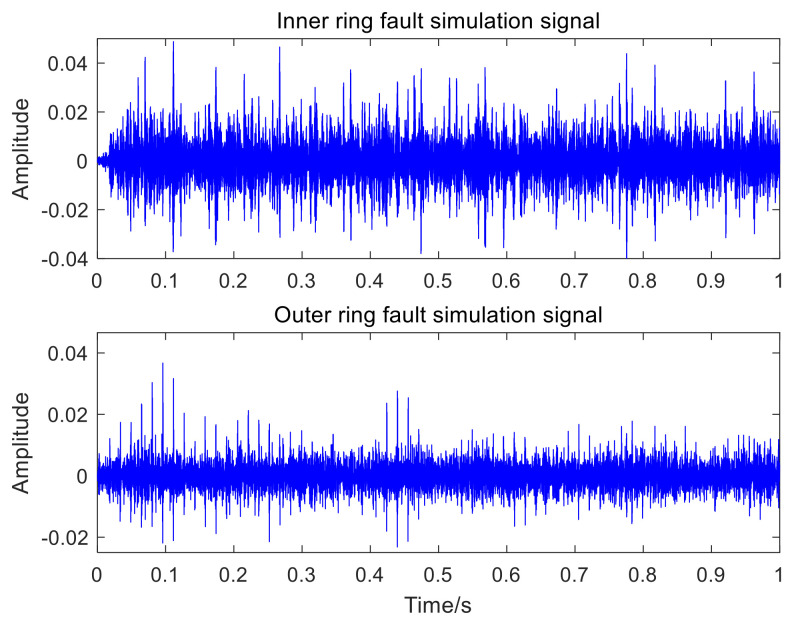
Time-domain waveform after MCKD processing.

**Figure 7 sensors-22-06769-f007:**
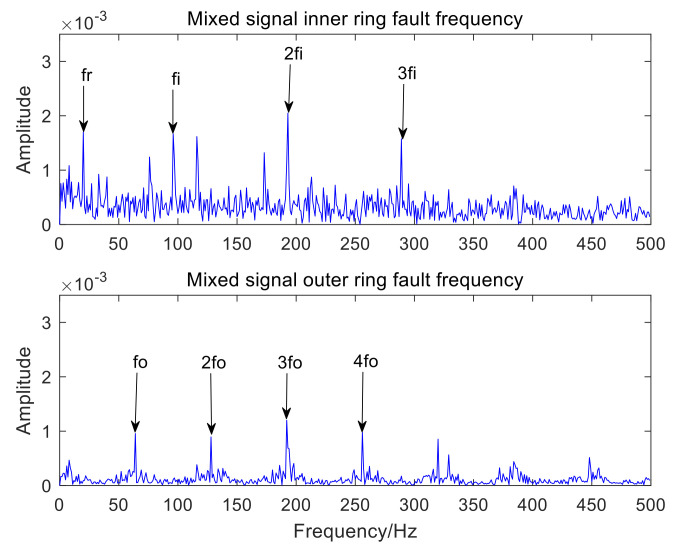
Envelope spectrum results using the AVMD-IMVO-MCKD.

**Figure 8 sensors-22-06769-f008:**
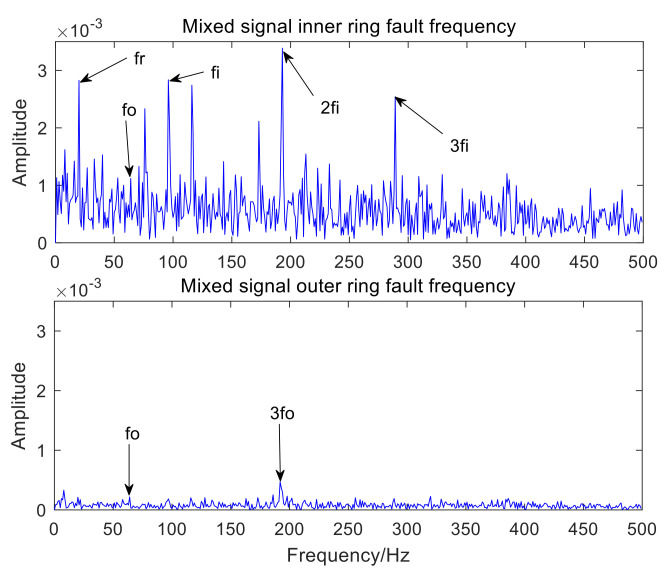
Envelope spectrum results using the KL-VMD-MVO-MCKD.

**Figure 9 sensors-22-06769-f009:**
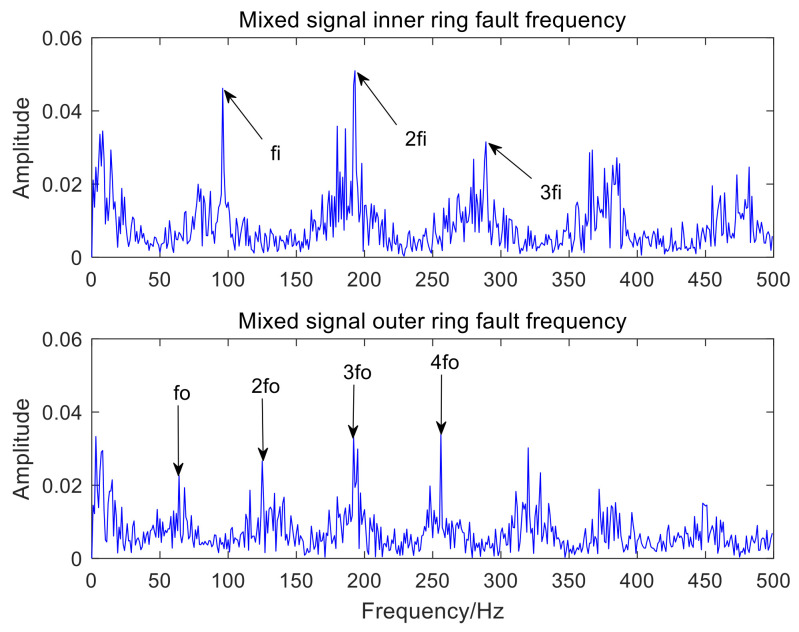
Envelope spectrum results using the EEMD-MVO-MCKD.

**Figure 10 sensors-22-06769-f010:**
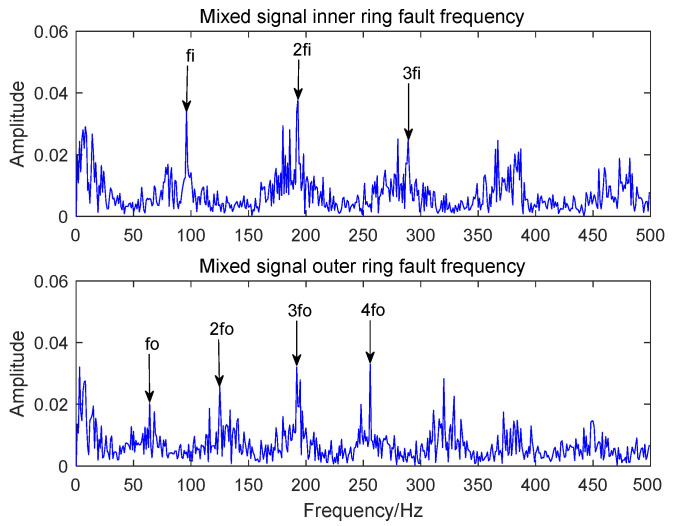
Envelope spectrum results using the ICEEMDAN-MVO-MCKD.

**Figure 11 sensors-22-06769-f011:**
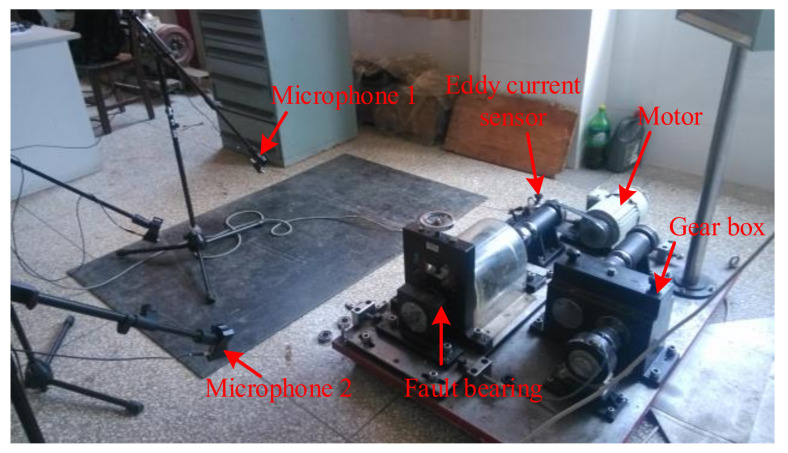
The layout of the test bench and microphone.

**Figure 12 sensors-22-06769-f012:**
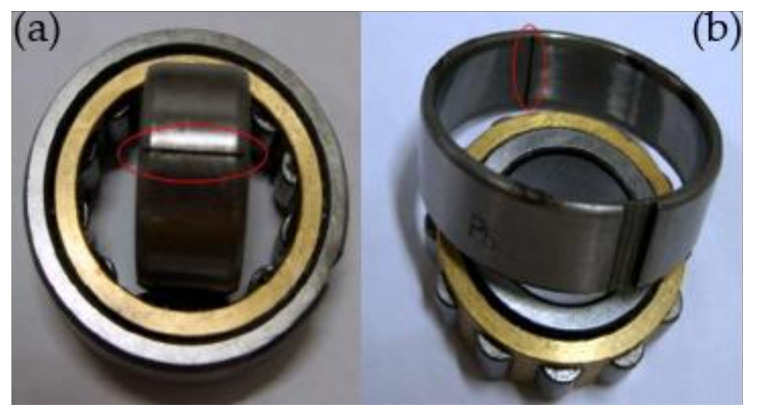
Type of rolling bearing failure. (**a**) is the inner-ring defect, (**b**) is the outer-ring defect.

**Figure 13 sensors-22-06769-f013:**
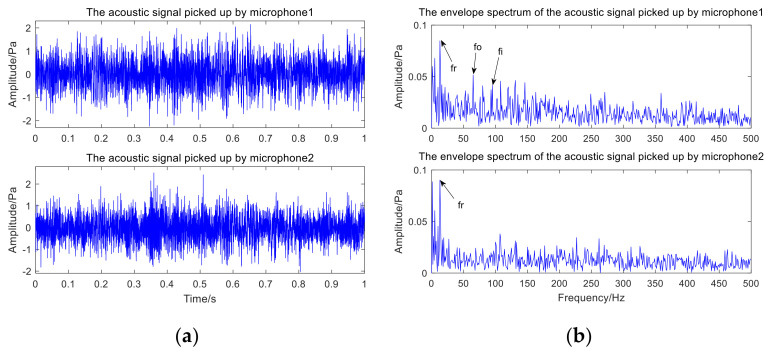
Time and spectral domain results of compound fault acoustic signals: (**a**) time-domain waveform of acoustic signals; (**b**) envelope spectrum of acoustic signals.

**Figure 14 sensors-22-06769-f014:**
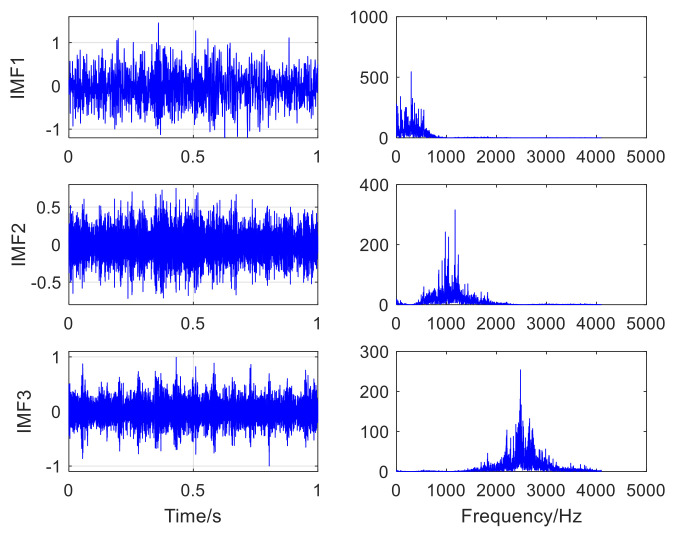
The result of the acoustic signal is decomposed by AVMD.

**Figure 15 sensors-22-06769-f015:**
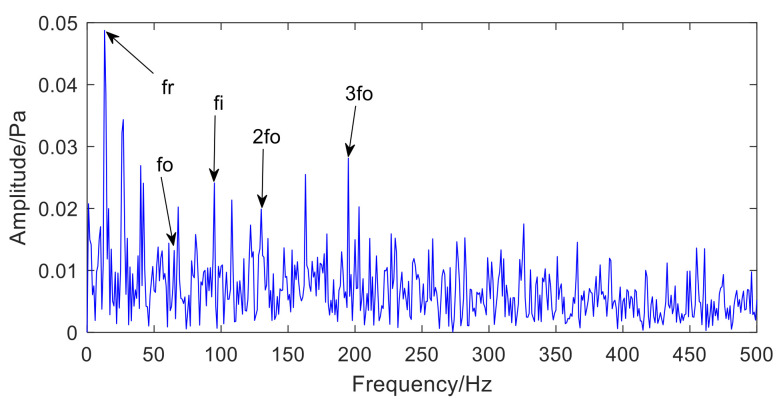
The envelope spectrum of the optimal component IMF3 selected by information.

**Figure 16 sensors-22-06769-f016:**
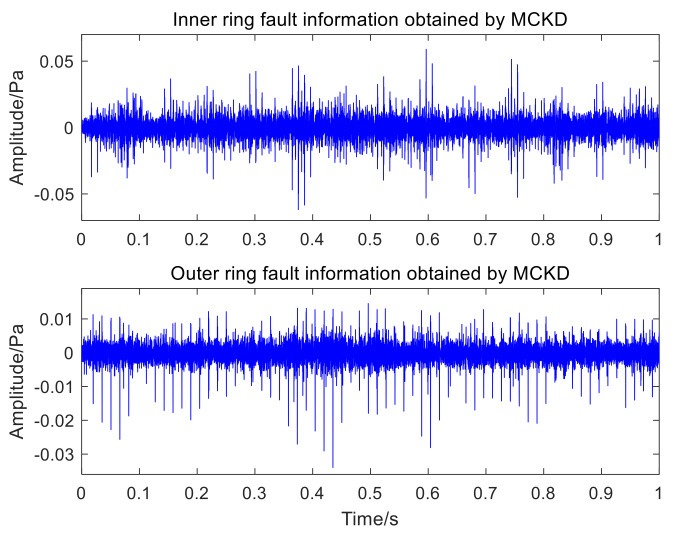
The time-domain waveform of an acoustic signal processed by MCKD.

**Figure 17 sensors-22-06769-f017:**
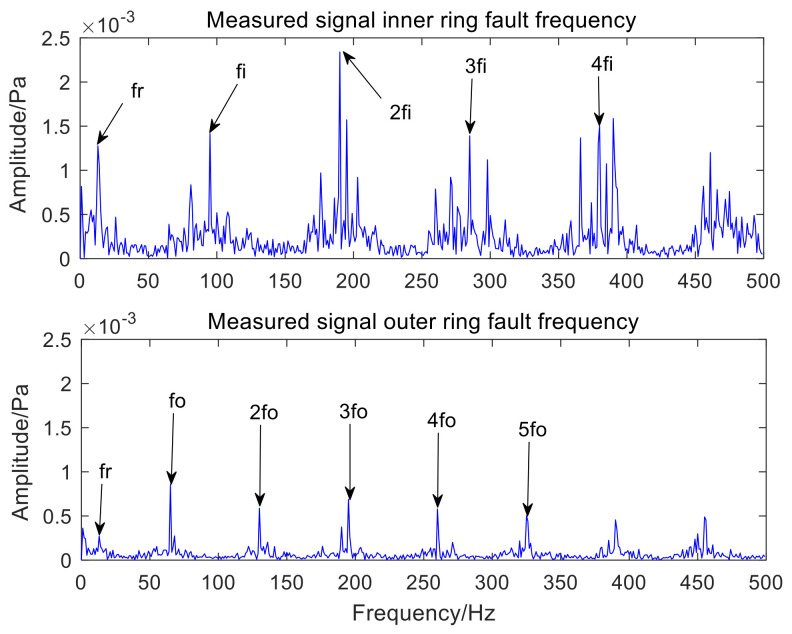
The envelope spectrum of an acoustic signal using the AVMD-IMVO-MCKD.

**Figure 18 sensors-22-06769-f018:**
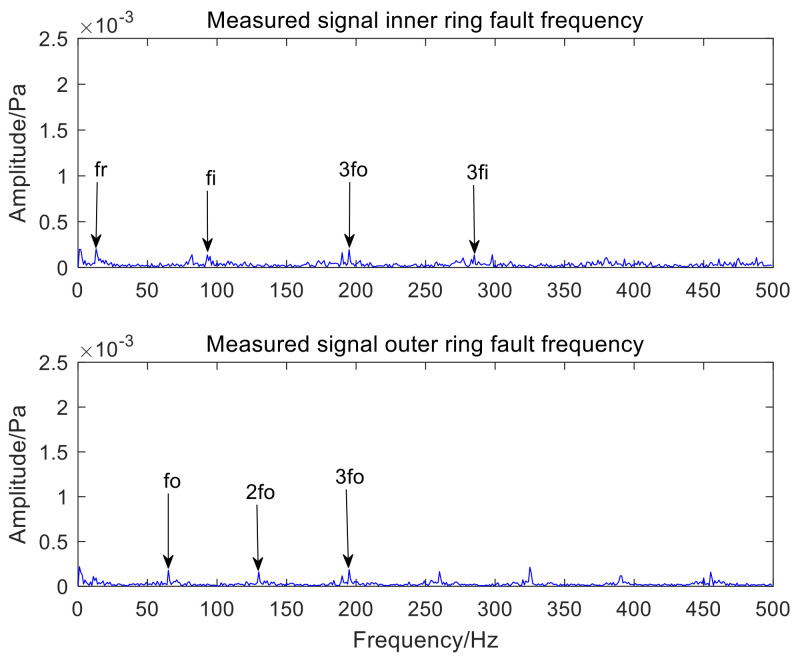
The envelope spectrum of an acoustic signal using the KL-VMD-MVO-MCKD.

**Figure 19 sensors-22-06769-f019:**
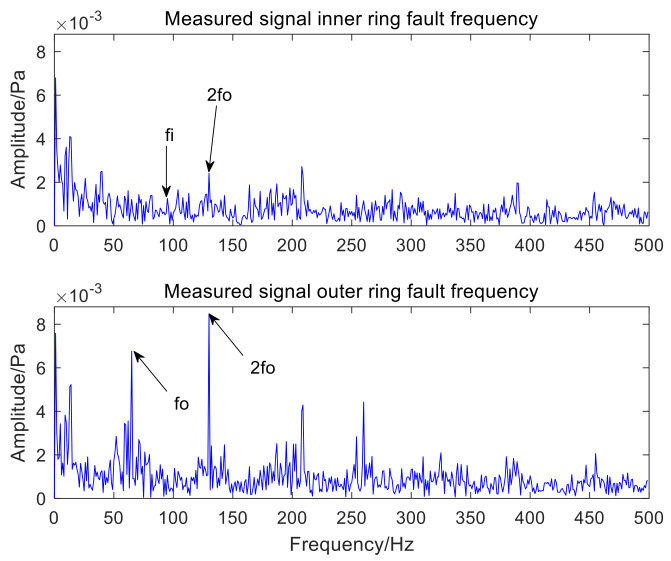
The envelope spectrum of an acoustic signal using the EEMD-MVO-MCKD.

**Figure 20 sensors-22-06769-f020:**
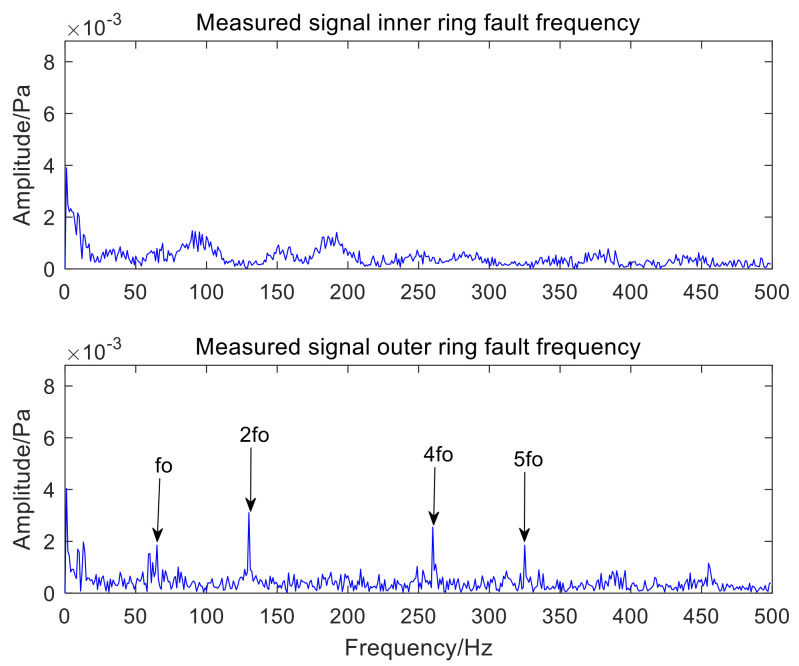
The envelope spectrum of an acoustic signal using the ICEEMDAN-MVO-MCKD.

**Table 1 sensors-22-06769-t001:** Simulation signal parameters.

Parameters	Values
Rotational frequency $f_{r}$/Hz	20
Inner ring amplitude $A_{0}$	3
Outer ring amplitude $A_{1}$	4
Inner ring fault frequency/Hz	95
Outer ring fault frequency/Hz	64

**Table 2 sensors-22-06769-t002:** Information entropy of each IMF component.

IMF	IMF1	IMF2	IMF3	IMF4
Information entropy	2.2819	2.2349	1.8184	2.1274

**Table 3 sensors-22-06769-t003:** Bearing parameter model type.

Parameters	Values
Type	NU205
Fault size	15 mm × 0.5 mm × 0.5 mm
Pitch diameter	39 mm
Diameter of cylindrical roller	7.5 mm
Number of cylindrical rollers	12
The contact Angle	0
RPM	800 r/min

**Table 4 sensors-22-06769-t004:** Information entropy of each IMF obtained after AVMD decomposes the acoustic signal.

IMF	IMF1	IMF2	IMF3
Information entropy	2.1683	2.2323	1.8919

## Data Availability

Not applicable.

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
