# Peer review of "Compound Fault Feature Extraction of Rolling Bearing Acoustic Signals Based on AVMD-IMVO-MCKD"

_sensors, 2022, doi:10.3390/s22186769_

Round 1
Reviewer 1 Report
Here are some suggestions and comments for the authors to address before the paper is accepted.
1. Provide author's name for reference in the introduction section instead of using “Reference [10,11] combined..”, “Reference [12] used”, and others.
2. Authors should explain major novelty and contributions of this paper in the introduction section. What are the advantages of the proposed work comparing to the existing methods?
3. Since Section 2 just simply explains VMD and MCKD, it can be combined with the section of proposed method.
4. Please include explanation and discussion of Fig. 1 in the content.
5. For the serial optimization scheme in Fig1, K is optimized first and then the alpha. However, is K guaranteed to be optimal with the new alpha? In other words, K and alpha should be optimized concurrently.
6. What does r_4 represent in Eq.(7)? Please provide more explanations on these random values r_1 to r_4
Reviewer 2 Report
In this paper is proposed applayed adaptively select the parameters of VMD and MCKD, on basis adaptive optimization method of VMD. An improved multiverse optimization (IMVO) algorithm is used to determine the parameters of MCKD.
The presented model enables the determination of bearing damage both in the low-frequency and in the high-frequency domain. Which is very important for determining early bearing failure.
One of the disadvantages of the proposed method is that the method is difficult to apply in an online environment, but it is not impossible.
After checking the English language and style, the paper should be accepted for publication.
Author Response
Thank you for your valuable comment and recognition of our work. The comments and suggestions are all valuable and very helpful for revising and improving our paper, as well as the important guiding significance to our researches。After checking the English language and style of the paper, we have finished the revision. The revised portions were marked in red in the revised manuscript. Thank you for your useful suggestion.Finally, if any further changes are needed, the authors are willing to fulfill them.
Reviewer 3 Report
This manuscript addressed compound fault feature extraction of rolling bearing acoustic signals. A fault feature extraction approach has been proposed which combined adaptive variational modal decomposition (AVMD) and improved multiverse optimization (IMVO) algorithm parameterized maximum correlated kurtosis deconvolution (MCKD), named AVMD-IMVO-MCKD. In general, this work is meaningful and valuable. To improve the quality of the manuscript, suggestions are listed as follows.
(1) The contributions of this work are barely highlighted.
(2) In terms of the comparison result, the author used EEMD-MCKD and ICEEMDAN-MCKD as comparative methods in order to illustrate the effectiveness of the proposed method (AVMD-IMVO-MCKD). However, this comparison can only illustrate the difference between EEMD and VMD as these three methods are all based on the MCKD. Please indicate the advantages of the proposed AVMD-IMVO compared with the appropriate VMD-based methods.
(3) In section 3.1, the author determined the parameters of the VMD according to the minimum IRE. What is the theoretical basis of IRE? Could the author explain the advantages of the minimum IRE for selecting the optimal parameters of the VMD?
(4) Please check the consistency of the words. for example, ICEEMDAN-MCKD or ICEEMDANMCKD.
(5) The analysis of the result is week. The author should quantitatively explain the difference of the proposed method with two comparison methods.
(6) The ordinate scale is inconsistent.
(7) English language is acceptable in general, but there are some grammatical errors that should be corrected. It should thoroughly proof read once.
From this point of view, I agree to accept the paper with major revision to publish in the journal as a research paper.
Round 2
Reviewer 1 Report
Authors have addressed my comments
Author Response
Thank you for your recognition of our work.
Reviewer 3 Report
The author addressed all reviewers' concerns and the revised manuscript satisfied my comments except the No.6 comment- The ordinate scale is inconsistent.
The ordinate scale of all the figures 7-10,16-19 should be consistent.
